# Dialogue between the Concept of the Object in the Theater of Tadeusz Kantor and the Theatrical Praxis of the Periférico de Objetos

Katarzyna Cytlak

Department of the History of Modern and Non-European Art, University Nicolaus Copernicus, 87-100 Toruń, Poland; cytlak@umk.pl

**Abstract:** Tadeusz Kantor was a Polish artist and theater director who directly influenced the conceptual understanding of theater, especially in Argentina following two visits to Buenos Aires with his troupe Cricot 2 in the 1980s. He exerted a particularly strong influence on the Periférico de Objetos [The Periphery of Objects], a troupe founded in Buenos Aires in 1989 by Daniel Veronese, Ana Alvarado and Emilio García Wehbi, labelled by critics as "the Argentine theatre of the image". Despite radically different socio-cultural contexts, elements arising from Kantor's theater practices (especially his idea of the "poor object" and his concept of "reality of the lowest rank") acquired distinctly different meanings in Latin America from those coined by Kantor. A nuanced examination of the Periférico de Objetos indicates that Kantor's concepts, which in their original context resisted politicization, played an important role in the creation of a socially and politically engaged theatre. His concepts, adapted to local realities by the Periférico de Objetos, were reflected in debates surrounding the recent Argentinian past, most notably, the post-dictatorship period.

**Keywords:** Tadeusz Kantor; the Periférico de Objetos; socially engaged theatre; theatre in Argentina; theatre in Poland; post-dictatorship theatre

## 1. Dialogue between the Concept of the Object in the Theater of Tadeusz Kantor and the Theatrical Praxis of the Periférico de Objetos

Among the Eastern European artists whose work undoubtedly had an impact on Latin America, and more precisely, on Argentina, a key figure is the Polish painter, director and performance artist Tadeusz Kantor. His presence in Buenos Aires dates back to August 1965, when he took part in an exhibition with other Polish artists at the Instituto Di Tella—the main institution devoted to the promotion of contemporary art in the 1960s. The exhibition, Arte actual de Polonia (Current art from Poland), was organised at the Centro de Artes Visuales del Instituto Torcuato Di Tella, Buenos Aires, from 6 to 27 August 1965. At the event, Kantor exhibited one of his umbrella assemblages: *Paragua de Oslo (Umbrella from Oslo)* (1965, hessian 162 × 100 cm). The work was reminiscent of the combine paintings of the American artist Robert Rauschenberg, who incorporated into his work various photographs, illustrations and objects found on the street. Kantor's artwork was situated between painting and relief, as it included a painting, on top of which an umbrella was affixed. Two years later, in 1967, Kantor was honoured with a prize at the ninth São Paulo Biennial in Brazil for a similar composition—*Emballage II.*[1] The difference here was that the figure of a woman was painted on the lower part of the work.[2]

In June 1965, in issue 25 of the magazine *Caballete* (Easel), the section "Desde Alemania" (From Germany) featured an article entitled "Las relaciones entre el teatro y las artes plásticas" (Relations between theatre and visual arts). The article was about the exhibition "Bild und Bühne" (Image and Scene) (Mahlow 1965),[3] organised in Baden-Baden, West Germany, by the German art critic Dietrich Mahlow (*Caballete* 1965, p. 12). Among the more than 500 art objects, artists' sketches and models on display—and the participation of

set designers and theatre directors from Europe and Japan—there were a number of works from both Czechoslovakia and Poland. The article quoted a long passage from the German newspaper *Süddeutsche Zeitung* describing the specificity of experimental theatre in Eastern European countries, and specifically mentioned the theatrical activity of Tadeusz Kantor:

> The "Süddeutsche Zeitung" writes: "While in some Eastern Bloc countries (Poland and Czechoslovakia) theatre has received a strong evocative and suggestive impulse from painters and decorators whose ingenuity goes hand in hand with their imagination, in other countries no such development has taken place. The theatres of the Federal Republic occupy an intermediate position between the radicalism of the young, such as Tadeusz Kantor, who in his 'Manifesto for Zero Theatre' vehemently rejects all stylistic directions in current theatre, and the conservative point of view; the scenography is the 'ancilla dramiatis', fulfilling only an intermediary mission"[4] (*Caballete* 1965, p. 12)

This was the first reference to Tadeusz Kantor's stage activity in Argentina. The article mentioned his *Manifest teatru zerowego (Manifesto For Zero Theatre)* from 1963, in which the Polish director expressed his interest in a theatre without stage action, based on the "disinterested states"[5] of the human being, such as "aversion, loss of will, apathy, boredom, monotony"[6] (Kantor 2000, p. 268).[7] However, recognition of Kantor's theatrical work did not reach Argentina until the 1980s and was mainly focused on his most mature theatrical activity, dating from that time.

Kantor travelled to Argentina with his theatre troupe Cricot 2 twice, performing his plays at the Teatro Municipal General San Martín in Buenos Aires. *Wielopole, Wielopole* was staged from 19 to 30 September 1984 and *Niech szczezną artysci!* (*Let the artists die!—¡Qué revienten los artistas!*) from 20 to 30 August 1987. Both stage performances were part of his Latin American tournée, which also included other Latin American countries.[8] In parallel to the presentation of his plays, Kantor gave lectures about his theatre and his work, and came into contact with the Argentinian public.[9]

As Argentinian theoreticians of the theatre and researchers such as Carlos Fos have stated,[10] Kantor's presence was important for Argentine theatre, as both visits gave impetus to a kind of "stage revolution".[11] In a recent publication summarising fifty years of the Teatro San Martín's activity we read:

> The visit of Tadeusz Kantor and his Teatr Cricot 2 from Krakow is, together with that of Pina Bausch, the most transcendental of all those the San Martín has received in its history. And the show he presented, Wielopole Wielopole, a truly singular aesthetic and human experience, whose influence on the Buenos Aires stage would leave a persistent mark[12] (Saavedra 2010, pp. 166, 184).

The importance of Tadeusz Kantor in the context of Buenos Aires theatre is reaffirmed by the words of researcher Julia Elena Sagaseta in her article "El teatro de imagen" (The Theatre of the Image):

> The work of Tadeusz Kantor, the Polish director and visual artist, is, for many filmmakers of this tendency [theatre of the image], the paradigm. The works of this director that were performed in Buenos Aires, Wielopole Wielopole and Let the Artists Die!, as well as his writings collected in the book The Theatre of Death, made a strong impression on some actors and directors who saw in them a path to follow. Other artists followed a vague line between theatre and performance[13] (Sagaseta 1992, p. 73).[14]

Among the many Argentinian groups in dialogue with Kantor, the most important was the Periférico de Objetos (Periphery of the Objects), founded in 1989 by Ana Alvarado, Emilio García Wehbi, Alejandro Tantanián, Román Lamas, Daniel Veronese, Felicitas Luna and Javier Swedzky (Propato 2002, pp. 285–96). This group—specialising in puppet theatre—acknowledged Kantor as its most important influence and spoke openly of the importance of the Polish artist's ideas to their work.[15] The origin of this relationship would have been direct contact with the author, since according to Ana Alvarado, the members of the group would have participated in the lectures Kantor gave in the country and in the debates following his presentations.[16]

## 2. The Concept of the *Poor Object*

Generally speaking, the reception of Kantor's theory of the Periférico de Objetos concentrated on the concepts of reality of a lower-rank (degraded reality) and poor object, developed by Kantor in his *Ambalaże manifest* (*Emballage Manifesto*) of February 1964 (Kantor 1964; in French in Bablet 1977, pp. 69–80).

The manifesto can be seen as a consequence of his "loss of confidence in painting" and his introduction to the plastic creation of banal, frequently used everyday objects, which the artist described as "poor" (Kantor 1964).[17] Although removed from their everyday reality and transferred to the context of art, Kantor's objects were not "found objects" introduced in a purely accidental manner like Dadaist objects.[18] The artist formed his own repertoire of objects from which he selected those that appeared in his artistic creations and in his theatrical works. He introduced cartwheels, folding chairs, coat hangers, umbrellas, suitcases, rucksacks, etc.

The objects that the artist used in his visual work and in his theatrical pieces had an ordinary, simple and popular character, as they were present in everyone's life and easily recognisable. Once selected, or rather added by Kantor to his repertoire, they did not lose their original characteristics. What differentiated them from their conventional versions was the extraordinary use he made of them, which reversed their usual perception and exploited their narrative, expressive and poetic side. This artistic appropriation of objects was not aimed at elevating quotidian objects to the status of art, but instead privileged the "descent of the artist into reality" and the "renunciation of his aspirations and his prerogatives of superior rank" (Borowski 1982, p. 18). Kantor called this process "creation around the zero point", a process whose purpose was not the repetition of the object in the field of art, but its "rescue" (*odzyskanie*) through creation (Borowski 1982, p. 18). These rescued objects became "actors" in happenings and theatrical performances, and parts of works of art such as collage paintings. Kantor defined the poor object as:

> . . .the simplest object, with marks of wear and tear, tarnished by long use, on the threshold of rubbish. Already for this reason [the object was]: useless in life, with no hope of fulfilling its vital functions. Of no practical value. Old junk! Simply: poor! Pitiful.![19] (Kantor 2005b, p. 415)

The concept of the object developed by the Argentinian group the Periférico de Objetos in the 1990s is, in many ways, reminiscent of Kantor's use of these common objects in his works. It is relevant at this point to compare Kantor's words on the poor object with the views of the members of the Argentinian group on the use of everyday objects on stage. According to Ana Alvarado, the object used in Periférico's plays should be: "a real, physical, artificial, irrational, found, constructed, disturbed or interpreted object [that] is subjected to an action"[20] (Alvarado 2009, p. 47).[21] In an interview in 1995, Ana Alvarado and Daniel Veronese expressed their affinities with Kantor's vision of the poor object, stating: "We are interested in incorporating into the scene the object taken and not modified, used artistically. The natural object brings a certain genuineness"[22] (Castillo 1995, p. 63). These words indicate that the group was not only familiar with Kantor's concept of the poor object, but also used objects associated with everyday life in their performances in a similar way. In this sense, we could say that the main function of these objects was, as in the case of the poor objects in Kantor's plays, to give a certain authenticity to the spectacle, or in the words of Alvarado and Veronese, a "genuineness".

In his analysis of the creation of the reality effect[23] in the literary works of Gustave Flaubert, the French realist novelist who wrote in the latter half of the 19th century, the French writer and semiologist Roland Barthes stated that the parts of the literary story that contain a description of the tiniest "concrete details"[24] are "taken immediately"[25] from everyday life and are "unnecessary"[26] or "superfluous in comparison with the [semiotic] structure"[27] of the story and the narrative, and primarily serve as a form of authentication for the narrative: they are what make it "plausible"[28] (Barthes et al. 1982, p. 81). Transferring Barthes' theory to the context of theatre, we can say the same about the introduction of

the real object, which not only contributes to the abolition of the boundaries between art and life, but also authenticates and contextualises the stage work. Both Kantor and the Periférico de Objetos bring their works closer to the everyday life of both the artists and the audience. Like Flaubert's details described by Barthes as insignificant notes outside the narrative (Barthes et al. 1982, p. 81), the objects used by these artists are included in their theatrical works not only for their formal aspect, but to give rise to a statement: "we are the real"[29] (Barthes et al. 1982, p. 81).

The fact that the objects the artists use in their performances are poor, ordinary, unsophisticated and invisible in everyday life reflects their desire to renounce all forms of illusion and seek authenticity in their works. It is a strategy that links them to the artistic practice and theory of early avant-garde creators such as Marcel Duchamp, who reappropriated everyday objects in order to inscribe them, without any formal intervention, in the world of art, thereby totally transforming their meaning without changing their appearance. However, even if the object in both Kantor and the Periférico de Objetos' work loses its original function, it does not become totally independent and free of its primary context, as in the case of Marcel Duchamp's ready-mades. The object "rescued" by Kantor and used by the Argentinian group maintains its characteristics and sense of origin. It seems to carry the memory of that original context, which it synthesises and expounds.

### 3. The Living Object, the Hybrid Object

All objects in Tadeusz Kantor's theatre and art are already worn out; they have their own history and traces of their formal degradation are visible. The objects that Kantor manipulates and appropriates for his creations are often somewhat intimate objects that establish a relationship of proximity with both the actors and the spectator. Kantor reuses these objects in a new context in order to bring out their dramatic character and their capacity to evoke emotions. In this way, inanimate objects are humanised in Kantor's works. In this sense, the objects in Kantor's theatre have several points in common with the object–subjects (*objets-sujets*) described in 1957 by the French philosopher Gaston Bachelard in his book *The Poetics of Space.* According to Bachelard, everyday domestic objects, such as a cupboard, a desk or a chest, are "the true organs of the secret psychological life [...] [which] have, like us, by us, for us, an intimacy"[30] (Bachelard [1957] 2004, p. 83). Kantor does not cite Bachelard's theories directly, but we can assume that during his stay in Paris[31] he may have come in contact with the French philosopher's writings. The artist seems to share with him his vision of objects which are humanised through their mental, emotional and also poetic potential.

Over time, objects gain a very particular meaning in Kantor's theatre. They cease to occupy the role of common props and gain a status similar to that of actors. The artist himself expands on this development when he states: "It is not that I want to make the object an actor: I want to make the object live, so that the object is like a living organism"[32] (Kantor 2003, p. 74). This phrase relates to a statement made by Ana Alvarado in 2009 in which the artist defines the specificity of the status of the object in the works of the Periférico de Objetos and its relation to the actor–manipulator that animates it. According to Alvarado:

In contemporary object theatre, the object does not replace the actor [...] The object must be on stage when its living reality is significant, when its presence structures not only the form but also the meaning of the work[33] (Alvarado 2009, p. 47).

In both cases, the object occupies a special place. In Kantor's works, as in the shows by the Periférico de Objetos, the objects present on stage are distinguished by being strongly marked by subjectivity. Not only do they acquire personal characteristics, but in both cases, they also act.

Kantor used everyday objects to create a kind of hybrid machine that acted in his plays. They were used to reduce the prominence of psychological processes within the performance and to restrict the actors' stage presence. The Polish artist thus introduced various hybrid objects such as: the mouse catcher/hospital bed, the camera/machine gun,

the toilet/cradle, the funeral machine or the coffee grinder/meat grinder. This practice of hybridising objects was also carried out with the actors' bodies. The relationship and, to a certain extent, the equivalence between the inanimate object and the human being also fascinated Kantor in his artistic activity. This is attested to by his drawings, in which he amalgamates the human figure with the everyday object: *The Man-Package* from 1963, *The Human Silhouette-Window* from 1971 or *The Human Silhouette-Tobot* from 1971.[34] Later, he also brought to life hybrid characters composed of actors' bodies attached to backpacks, armchairs or mannequins on stage. These played a leading role in *The Dead Class*, one of Kantor's most famous and evocative plays, first performed in 1975. Kantor's practice of hybridising objects with actors was well known to the members of the Periférico de Objetos. As proof, we can cite a text by Ana Alvarado in which the artist refers to her stage work and her conception of the hybrid object:

Kantor's characters are neither alive nor dead. Ghosts from the past or inhabitants of Kantor's own dreams are summoned onto the stage by Kantor himself. With their quasi-life they incessantly repeat something or repeatedly enter and leave the stage. Sometimes they are mistaken for mannequins or other objects in the scene, sometimes they are attached to the objects they define: Woman behind the Window (She herself carries the window through which she looks). She only exists in concrete form because she is attached to the window. Kantor's cast of actors is composed of "eternal wanderers", the bodies and costumes, multi-layered "packaging", are welded, glued or sewn to suitcases, bicycles, windows, guillotines[35] (Alvarado 2009, p. 40).

In her text, Alvarado refers to one of the characters in the spectacle *The Dead Class* from 1975, represented by a woman amalgamated with a window. Kantor's ideas about personalised and hybrid objects allowed the Argentinian group to conceptualise, in their works, the relationships between the object, the actor–manipulator who moves the object and "the non-manipulating actor"[36] who "links with the object without becoming it"[37] (Alvarado 2009, p. 60). In the same way as Kantor, who introduced mannequins into his theatre in 1967,[38] the Periférico de Objetos used "anthropomorphic objects"[39]—all kinds of dolls that coexisted alongside the actors and manipulators on stage. In fact, Alvarado very precisely defines the types of relations possible between the three "scenic subjects"[40] and very clearly differentiates between their capacities and their limitations. According to her, in today's object theatre the manipulator "acts on [the] object in such a way that it is not possible to be sure where one ends and the other begins"[41] (Alvarado 2009, p. 47). Similarly, in the notes on his play *Wielopole, Wielopole* from 1980, staged in Buenos Aires in 1984, Kantor stated:

The matter of the show was created by the 'inner life' of the OBJECT, its characteristics, its destiny, its imaginary area. The actors became its living parts, its organs. They were as if genetically united with that object. They created a certain BIO-OBJECT, which acted and secreted the plot of a rather special action.[42] (Kantor 1984b, p. 133).

In both cases, the object occupies the leading role in the action, but, as Alvarado argues, the manipulator can "also be 'the other' who is embodied in her human body"[43] (Alvarado 2009, p. 47). In this case, the artist conceives the presence of a third actor, defined by her as a "non-manipulator"[44], whose presence in the scene comes from a place that is rediscovered as that of a non-object.

In order to better understand Kantor's intentions, we can refer to Alfred Gell's theory on the agency of objects (Gell 1998). According to Gell, the latter have performative abilities very similar to living beings and can substitute for them. Kantor's objects, which are also artworks, have the power to influence their viewers, to communicate with them and to provoke "social-emotional responses" (Gell 1998, p. 6). By using ordinary and worn object agents, Kantor aimed to create a mental state—an ambience of hominess familiar to the spectators. The reason for this was to penetrate deeply inside their lives, to affect them, to "enmesh" them "in a texture of social relationship" (Gell 1998, p. 17). Similarly, in the plays of the Periférico de Objetos, objects are representations of human beings and are capable of social agency. Their aim is similar to that of Kantor's objects: effectively engaging the

spectator and creating an atmosphere that is both familiar and, in some way, also strange to him in order to deeply penetrate him affectively.

## 4. Objects in Context

What is particularly meaningful in the dialogue surrounding Kantor's theatre is that the works of the Polish director acquired a completely new meaning in the Argentinian context. In the context of the People's Republic of Poland, under a communist regime, Tadeusz Kantor was among the artists who refrained from publicly expressing his opinions on the policies of the authorities. It is known that after World War II he spoke out against the doctrine of social realism and, as a consequence, lost his job at the art school in Krakow in 1949, but then, during the following decades, he was one of the few Polish artists permitted to travel and exhibit abroad.[45] At that time, Kantor simply refused, more or less explicitly, to speak out through his works about the current politics of his country. On many occasions, he stated that his only desire was to question the limits of art. Thus, experimentation was the only important thing in his artistic oeuvre, happenings and theatrical work. However, his highly metaphorical art and stage productions could be read, and certainly were, as allegories full of allusions to state oppression. As an example, we could mention his show entitled *Szafa–Der Schrank* (Wardrobe), an adaptation of the play *W małym dworku* (In the Little Manor House) by the Polish writer Stanisław Ignacy Witkiewicz, written in 1921.[46] The play was first shown in Baden-Baden, West Germany, in 1966, and it was interpreted as a metaphor for state oppression and the enclosure of the individual in communist countries.[47] However, Kantor, unlike the members of the Periférico de Objetos, refused to openly criticise the socio-political reality of his time.

During his stay in Buenos Aires in 1984, Kantor was questioned about the political reality of his country. Asked whether it had been possible to perform *Wielopole, Wielopole* in Poland in the context of martial law, Kantor replied:

Yes, on two occasions. The first was in 1980, when the Solidarity movement was emerging. In that sense, I think my play predicted everything that happened with Lech Wałęsa's revolution and the war that came afterwards. At that time, we did performances in the Gdańsk shipyards, where the Solidarity trade union started. The second time was in December 1983, in the middle of the state of siege.[48] (Di Tella 1984).

Kantor's opinion was openly against the idea of militant theatre. When asked about the reception of his scenic plays in Poland in the context of martial law in the early 1980s, he stated:

The reaction of the audience was very special. The temperature was higher; but I don't know if that's a good thing because I don't like it too much when a work of art provokes a reaction in which its politics raises the temperature. It is the work of art that should create a special temperature. [...] Actually, I am against political theatre[49] (Di Tella 1984).

In the same vein, the Polish director stressed that: "in Poland, at [that] time, [there were] many directors who [took] advantage of politics, but only to be successful, to make a career"[50] (Di Tella 1984).[51] He continued: "the theatre that [made] Cricot 2 [was] not political theatre, it [had] no political programme to [support] it. Even if, in its deepest layers, [it] had a political significance"[52] (Di Tella 1984).

Kantor's reluctance to take a stand against authoritarian power in his home country makes it all the more complex that, in Argentina, the performance of his plays, among others by the Periférico de Objetos, took on strong political overtones. The Periférico de Objetos was, in this sense, one of the groups that directly made power relations a primary theme in their plays. The first play staged by the group in April 1990, Alfred Jarry's *Ubu Roi* (Ubu King) from 1896, was a farce about authoritarian power. In the second show, *Variations on B... (Beckett),* performed in May of the same year, there was a torture scene performed by a puppeteer playing the role of a tyrant. The central element of the next show, *The Sandman* from April 1992, whose name was taken from a story written by E.T.A. Hoffmann[53] and belonged to the *Schwarze Romantik* (Dark Romanticism) literary genre, was a sandbox full of hidden dolls. The dolls became visible only to be covered again with

sand, manipulated by the actors who played widows in the story. Their gestures clearly evoked the practice of disappearance carried out as part of state terrorism in the country.

As the Argentine theatre historian Jorge Dubatti justly stated, the group's performances created a very dark, pessimistic atmosphere:

From a semantic point of view, Veronese constructs in Cámara Gesell [the fourth show produced by the Periférico de Objetos, in February 1994] an absolutely violent, pessimistic and negative image of man, he elaborates a nihilistic anthropology that takes on all the tragedies of the past (and of the foreseeable future) without concessions to conformism or hope[54] (Dubatti 1995, p. 32).

It is possible to affirm that the main difference between Kantor's theatre and that of the Periférico de Objetos, beyond the dissimilar theatrical forms (experimental theatre in the case of the former and puppet theatre, also experimental, in the case of the latter), consists of the frequent references by the Argentinian group to violent and aggressive behaviour. These can be associated with the repressive practices and violence generated by the military dictatorship in Argentina. To an extent, all the group's works can be interpreted as metaphors associated with the repressive mechanisms of authoritarian power. Tadeusz Kantor addressed the question of death in his theatre. His manifesto of the *Teatr Śmierci* (*Theatre of Death*) from 1975 is one of his best-known texts and was translated in Argentina in 1984. In fact, Kantor's two shows in Argentina belong to this last stage in his theatrical work, called the theatre of death, which begins with this manifesto.[55] Moreover, Kantor understood the "reality of a lower rank" as the part of everyday human life, linked to death, which is denied. In shows such as *¡Qué revienten los artistas! (Niech szczezną artysci!—Let the artists die!)*, performed in Buenos Aires in 1987, Kantor focuses on his own death, staging sequences of memories from his own life and prophesying his demise. However, violence, as such, was not a subject of particular interest to Kantor.

Perhaps one moment in Kantor's theatre that seems closest to the atmosphere created by the Periférico de Objetos, in part linked to representation of the mechanisms of repression, is his show *Wariat i zakonnica* (*The Madman and the Nun*), based on a libretto by Stanisław Ignacy Witkiewicz, staged in 1963. In it, Kantor introduced the "annihilation machine", constructed from numerous folding chairs. The "machine" competed with the actors, pushing them off the stage and not allowing them to speak their lines. In the manifesto *Teatr autonomiczny. Manifest teatru zerowego* (*Autonomous Theatre. Manifesto of the Zero Theatre*) of 1963 (Kantor 2000, pp. 233–43), the Polish director characterised this hybrid stage furniture as violent, crazy, nervous, ridiculous and monotonous, and claimed that it should act in an autocratic manner. The machine focused on delimiting the stage space, which was, according to Kantor, "reduced to a space close to zero"[56] and "so thin and pale that the actors [had] to struggle to stand on it"[57] (Kantor 2000, p. 238). The "annihilation machine" thus created a situation of competition and threat. As we will observe later in the performances of the Periférico de Objetos, in this work by Kantor, a banal everyday object inserted into a power structure becomes monstrous due to its unusual function.

Unlike the Periférico de Objetos, Kantor's theatre is based on his own personal memories, his intimate and highly subjective experiences. It is a form of play based on his own memory. The artist examines death from a dimension that is firstly personal and then universal, but is not a concrete collective experience. The Argentinian group renounces the fundamentally subjective character of Kantor's theatre in its work, and seems to take collective memory, as determined by its social contexts, which, in this case, relate to the recent history of Argentina, as the object of its investigation. The work of the Periférico de Objetos played a very central role in the post-dictatorship period in Argentina, acting as a catalyst for the discussion of traumatic experiences from its recent history that had previously been silenced. Its reception in the country was inseparably linked to a process of memory and mourning, both individual and collective.

### 5. The Sinister Object

As María Castillo states, the Periférico de Objetos "did not [leave] the spectator indifferent" but rather challenged them with a "disturbing aesthetic"[58] (Castillo 1995, p. 60). This is something that, once again, brought them close to Kantor, who was well known for his use of a variety of techniques to draw the spectator directly into the action. In plays performed by the Periférico de Objetos, we can observe the presence of strategies aimed at narrowing the distance from the audience. For example, Jorge Dubatti recalls that, in the play *Cámara Gesell,* the design of the space places the spectator in close physical proximity to the dramatic space; in this way, by leaving the "audience ominously glued to the most cruel and obscene scenes"[59] (Dubatti 1995, p. 35),[60] the Argentinian group went further, seeking to put them in a state of shock or, at the very least, produce discomfort. This aim was affirmed by Daniel Veronese, who in an interview with María Castillo on the subject of the aesthetics of the group's plays, said that it was:

> an aesthetics of the obscene, because one of the peripheral elements is obscenity: to show what people are not expecting to see, to make visible what should have been left out of the scene[61] (Castillo 1995, pp. 60–63)

For her part, Ana Alvarado refers to the Freudian term "uncanny" (unheimlich).[62] According to her, the group's third play, *El hombre de arena (The Sandman)*, from April 1992, in particular was "strongly based on the Freudian notion of the uncanny"[63] (Alvarado 2009, p. 66). The Periférico de Objetos worked with surprise in order to achieve an effect similar to the state described by Sigmund Freud with the term *das Unheimliche*, characterised by a commutation between the known and the unknown. We may also note that E.T.A. Hoffmann's short story, *The Sandman* (see Figures 1 and 2), on which the Periférico de Objetos' show was based, was also one of the examples analysed by Freud in his essay *Das Unheimliche* (Freud [1919] 1989). The German word *unheimlich* stands in opposition to the terms *heimlich* (domestic) and *heimisch* (native), both of which come from the word *das Heim*, meaning "home", or *der Heimat*, meaning "homeland". However, the uncanny does not simply mean something opposed to the domestic, something that might be unfamiliar or strange. For Freud, what is important is the linguistic proximity between the two terms *Heim* and *unheimlich*, because the latter, according to Freud, contains in itself *das Heim*, or that which is familiar. In this way, the uncanny becomes a factor that provokes a feeling of unease or even fear, caused by the cognitive dissonance between the recognition of a common and familiar phenomenon or object and the discovery of its uncomfortably strange and foreign side. The uncanny manifests itself, then, when the pleasure of the known becomes the displeasure of a terrifying and unfamiliar experience. Slavoj Žižek underscores this very point when he states that it is familiarity that constitutes the crucial element of the uncanny. In his 1989 book *Looking Awry*, in which he introduces Jacques Lacan's concepts through popular culture, Žižek states: "The most familiar things take on a dimension of the uncanny when one finds them in another place, that 'is not right'. And the thrill effect results precisely from the familiar, domestic character of what one finds in this 'Things'" (Žižek 1992, p. 145).

The fact that the Periférico de Objetos used dolls in their works imposed an uncanny atmosphere on the stage, as shown in Figure 3. The use of dolls provokes associations with childhood; although of course, in these works they were animated for an entirely different purpose than their primary use, which is child's play. This play serves as a means of replicating the behavioural patterns of adults, and thus prepare a child for its future life. The group also reproduced adult behaviour with the dolls, but instead of representing situations from family life, they represented acts of violence, linked to the reality of a social life subjected to authoritarianism.

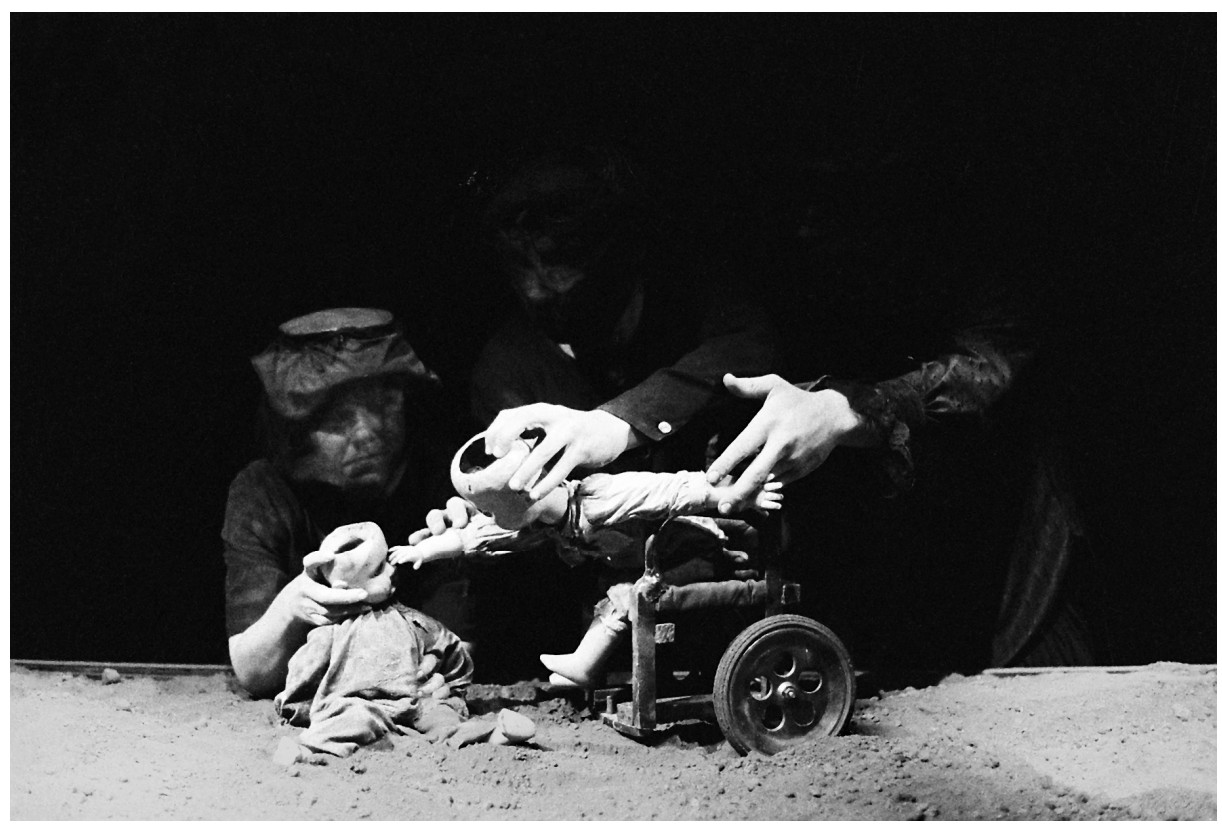

**Figure 1.** The Periférico de Objetos, "El Hombre de arena/The Sandman", Buenos Aires, 1992. Photograph by Magdalena Viggiani, Courtesy Archivo Ana Alvarado, Buenos Aires.

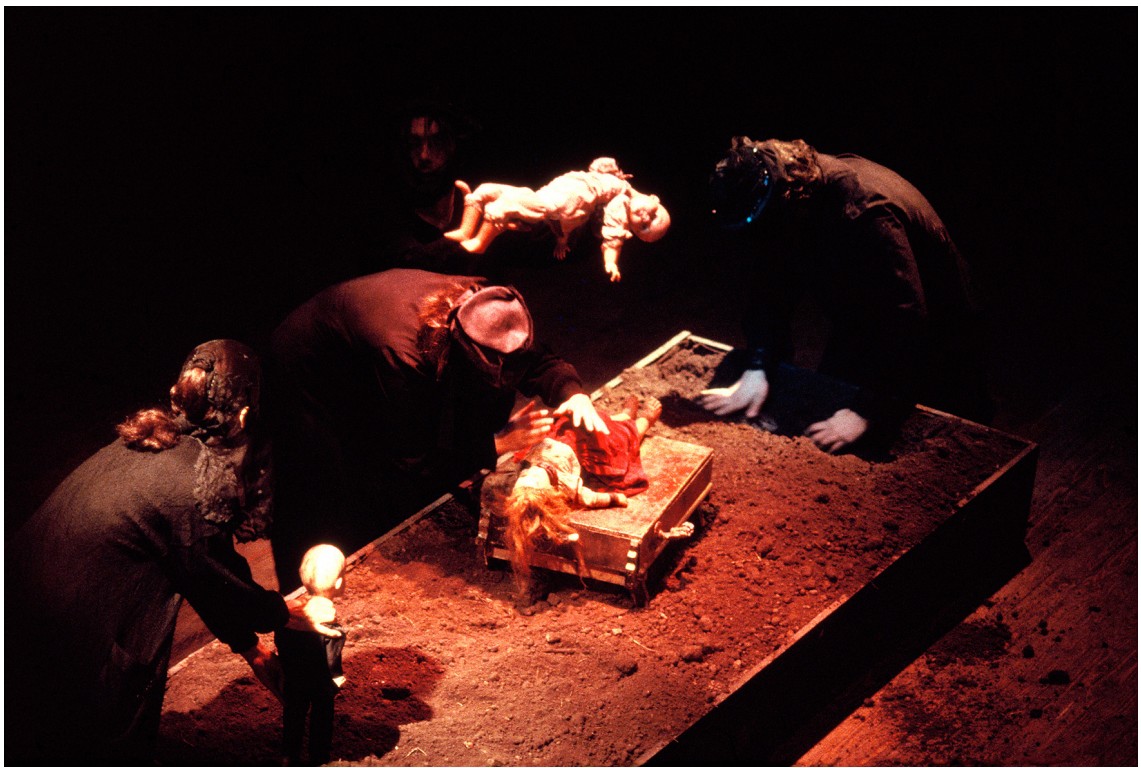

**Figure 2.** The Periférico de Objetos, "El Hombre de arena/The Sandman", Buenos Aires, 1992. Photograph by Magdalena Viggiani, Courtesy Archivo Ana Alvarado, Buenos Aires.

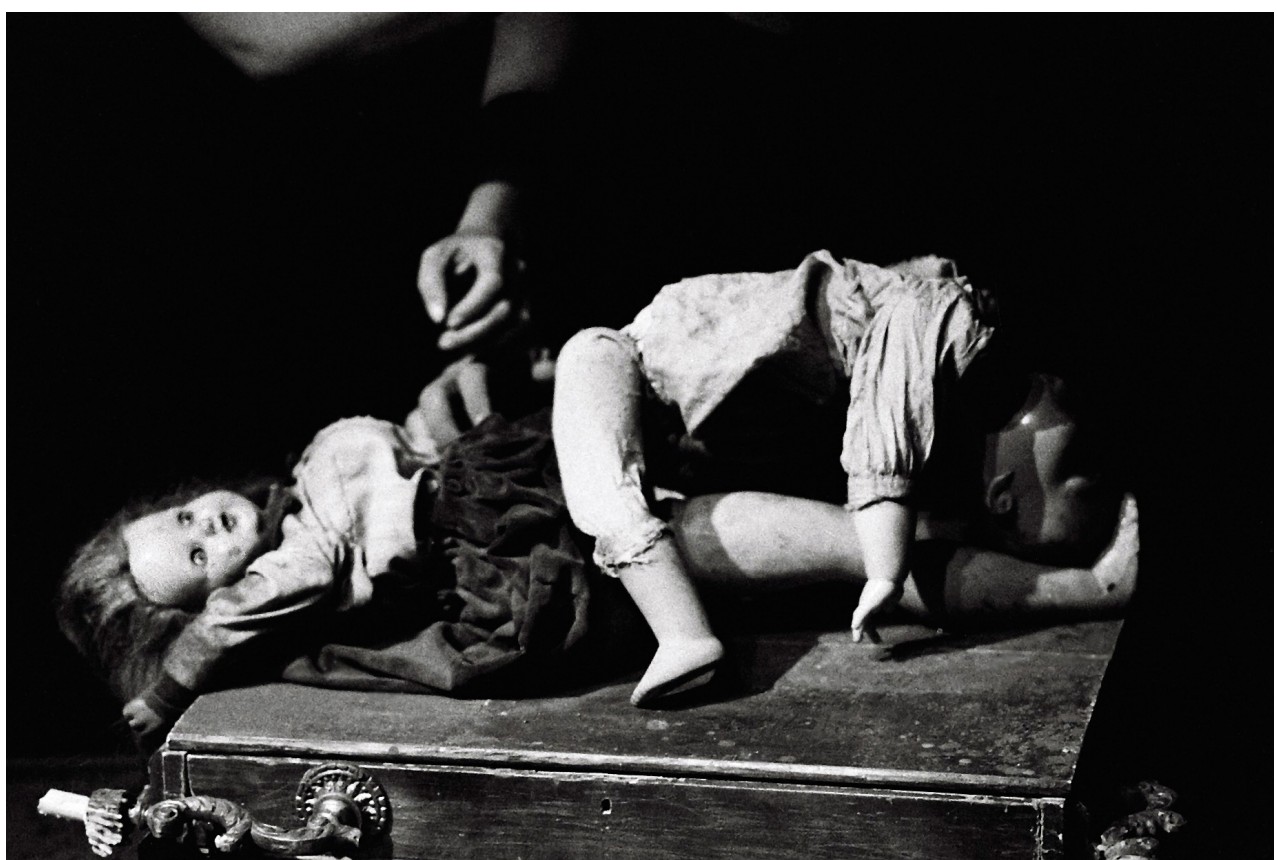

**Figure 3.** The Periférico de Objetos, "El Hombre de arena/The Sandman", Buenos Aires, 1992. Photograph by Magdalena Viggiani, Courtesy Archivo Ana Alvarado, Buenos Aires.

The familiar character of the objects used in the works, easily recognisable to the public as they could be part of their everyday life or have been part of their childhood, intensified their experience of discomfort. This discomfort was provoked by the association of these familiar objects with death. Through the use of such strategies, the works of the Peripheral Objects were transformed into powerful and disturbing spectacles.

The reception of the Periférico de Objetos's work is inseparably linked with its socio-political context. In this sense, we can cite the article "El elemento siniestro en las obras de El Periférico de Objetos" (The sinister element in the works of the Periférico de Objetos), by playwright and theatre critic Cecilia Propato, written in 1998 at the height of the group's stage activity. Propato states:

The disappeared are dead, and the sinister thing is to bring the issue to light. What happened is so sinister that society prefers it to be covered up, buried. "The periférico de objetos ", by bringing this subject to light, become obscene. They show what people don't want to see, and they can only do that by means of dolls. Here we come to the peripheral essence: the choice to work with dolls—which already have a sinister charge—is related to the possibility of telling horrible stories that could not be told with actors[64] (Propato 1998, pp. 387–95).

Indeed, the use of anthropomorphic objects allowed the group to depict scenes of violence in a very direct and explicit way. In fact, contrary to Propato's assertion, horrific stories linked to the last dictatorship were told in several films, with actors. The use of dolls in the works of the Periférico de Objetos allows this violent and sinister atmosphere to be maintained in a metaphorical dimension, which strengthens the effect of the work on the spectator. The link between the group's creation and the last Argentinian dictatorship is also perceived by Daniel Veronese. In his analysis of the construction process of the play *El hombre de arena* (*The Sandman*), the artist states:

Although there was a personal political need in all of us to exorcise certain themes related to military repression, this was not present when we prepared the work. But obviously in our country the work was read as a work about those who disappeared during the dictatorship. We, without having set out to talk specifically about the subject, achieved a poetic synthesis that I think would have been impossible to achieve if the idea had been ahead of the form. We simply let our ghosts wander over the sinister. And it is impossible that in Argentina certain signs are not read in that way; the sinister being an element with which we have lived for years. The result was not pamphletary. That was, I think, the greatest achievement of the work[65] (Veronese 2000).

Like Kantor, Veronese affirms the importance of the aesthetic value of creation over its political message. The staging of the sinister aspects of everyday life during the dictatorship emerges in the group's work in an unintentional way. The message of the work seems to emerge from the context in which the work is received. The inevitability of the association with the disappeared may then be related to a collective need to exorcise certain themes linked with the repression. Thus, the audience's level of disturbance in viewing the play is very high. Regarding the show *El hombre de arena*, Daniel Veronese refers to the two strong contradictory feelings that the play produced: disgust and attraction, both provoked by the return of the repressed. Recalling its impact, he states:

In the words of the audience, the result was terrifying. People couldn't bear to watch it, but at the same time they couldn't stop watching it. As if, despite the need to run out of the room, a force was keeping them attached to their seats. To be able to animate something that has no life produces that fascination, I think, that is so difficult to match. It's knowing something about death. The audience needs to see something about death even if they reject it[66] (Veronese 2000).

The disabled dolls used by the Periférico de Objetos, like Kantor's "dark parades of macabre puppets"[67] (Kantor 2005a, p. 33), are associated with the concept of the abject developed by Julia Kristeva. The abject occupies a liminal space, an intermediate place between the concept of object and the concept of subject, which disturbs the symbolic order and which she associated with both fear and pleasure.[68] The doll in the work of the Periférico de Objetos becomes an abject object because it simultaneously provokes discomfort and pleasure in the spectator. These feelings seem to intensify as the spectator realises that death is at work on the stage, and that the object, which was a toy, becomes a corpse. Kantor himself, in an interview in Buenos Aires with Andrés Di Tella in September 1984, analyses the role of dead people in his own theatre, stating that: "in the dead we can recognise ourselves, in the sense that the dead person is of our own species, he is a human being like us. But at the same time, he is a stranger, he is someone who is out of life"[69] (Di Tella 1984).[70] In his text on the *Theatre of Death*, Tadeusz Kantor said that introducing mannequins into his shows allowed him to address the theme of death. In this sense, he stated: "The MANNEQUIN in my theatre has become a MODEL, through which a strong sense of DEATH and the condition of the DEAD passes. A model for the LIVE ACTOR"[71] (Kantor 2005c, p. 18). For Kantor, the live actor should be able to internalise death from the model created by mannequin. His lifeless anthropomorphic figure embodies the presence of death on the stage. In a way, the Argentinian group appropriates the idea of the Theatre of Death to exorcise a sinister experience of Argentinian society.

Through the use of common, worn-out objects, bearers of their own history, Tadeusz Kantor and the Periférico de Objetos link themselves to the past. The old porcelain dolls are, for the members of the Argentinian group, like old photographs that "refer in a particular way to time and death"[72] (Castillo 1995, p. 63).[73] In this sense, the group comes very close to Kantor's consideration of mannequins. These anthropomorphic objects are used by the artist to express life through their absence. Kantor himself confirmed this phenomenon:

Its appearance [of the mannequin in my theatre] aligns with my conviction, which is growing stronger and stronger, that life can only be expressed in art by the absence of life, by reference to DEATH, by APPEARANCES, by EMPTINESS and the absence of TRANSFERENCE[74] (Kantor 2005c, p. 18).

Both theatrical visions are also inseparably linked to the context in which they were created. However, Kantor's degraded reality does not correspond exactly to the peripheries of the Argentinian troupe. Both Kantor and the Periférico de Objetos recreated an atmosphere in their plays that might suggest the structure of a dream or a memory. Both organise their stage performances in a way that is sometimes chaotic, illogical and inexplicable in the same ways that the memories that reside in human memory manifest themselves. Both play with trauma and the return to childhood: in Kantor's case, one could cite his play *The Dead Class*. However, unlike Kantor, who perpetually stages his own dreams, experiences, memories and anxieties, the performances of the Periférico de Objetos renounce this highly subjective character and take the plots of collective memory as the object of their investigation.

The post-dictatorial context becomes a crucial element in the analysis of the works of the Periférico de Objetos. This is confirmed by Ana Alvarado's self-analysis of the group's artistic activity. Referring to Jorge Dubatti's text "Micropoetics. Teatro y subjetividad en la escena de Buenos Aires (1983–2001)" (Micropoetics. Theatre and Subjectivity on the Buenos Aires Stage (1983–2001)) of 2002 (Dubatti 2002), Alvarado defines the origin of the stage works of both the Periférico de Objetos and other theatre troupes of the same period. She states:

In their works, [the theatre troupes of the post-dictatorship period in Argentina] refer not only to this stage of Argentinean history: Resistance, disappearance, violence, torture, repression, the sinister, etc., but also to the social and economic crisis of the subsequent period, which is addressed on stage in various ways: parody, absurdity, meaninglessness and pathos. These aspects are not "themes" or "influences" of the social context on the work, but are the building blocks of the work itself and the artists' procedures[75] (Alvarado 2009, p. 48).[76]

Alvarado points to the decisive link that the group maintained with the social and political context of Argentina in their artistic production. This context is related both to the need to persist in the practices of memory, referring to the last dictatorship at a time when the trials for crimes against humanity were in clear retreat, and to the social and economic crisis that characterised the 1990s.

However, the performances of the Periférico de Objetos cannot be treated as testimonies to the most painful events of recent history. They are not inspired by a specific biography, as in Kantor's case, but are based on the structure of a trauma which belongs to a collective. The nightmarishly recurring scenes of violence make the audience live and relive their own fears. The performances by the Periférico de Objetos tend to reproduce the mechanism of a bad memory which, once repressed, reappears in an undesirable way at the least favourable and least expected moments. The group focuses, then, on memories whose incessant return shatters the audience's comfort. In looking for the reasons why an image becomes intolerable, the French philosopher Jacques Rancière, in his analysis of representations of massacres, remarked that it is the fact that images are "too intolerably real to be proposed as images" (Rancière [2008] 2010, p. 85). Following this idea, we could affirm that the sinister or the intolerable in the work of the Periférico de Objetos is used first and foremost as a means of reworking a trauma linked to the experience of social oppression.[77] The antique doll serves, as the mannequin did for Kantor, to represent the unrepresentable, the intolerable. In this sense, the artistic work of the Argentinian group assumes a very important role in the post-dictatorship milieu as a means of supporting the discussion of recent history. Their work is inseparably linked to social debate about the crimes of state terrorism.

The work of the Periférico de Objetos is thus a very complex example of the dialogue between Kantor's art and theatre in Latin America. The Argentinian group found, in the Polish artist's scenic language, elements from which to develop its own language. Taking a position more marked by the political context, the Periférico de Objetos presents, unlike Kantor, a narrative that serves to relate to its spectators on the basis of a shared past. In any case, both artists' performances connect with universal archetypes. The fact that both

concepts of theatre are still captivating today is due to the fact that they speak, above all, about the human condition and its horrors.

**Funding:** This research received no external funding.

**Data Availability Statement:** Data available in a publicly accessible repository that does not issue DOIs.

**Conflicts of Interest:** The authors declare no conflict of interest.

## Notes

1    Tadeusz Kantor received, together with other new artists, a Prize at the IX São Paulo Biennial in 1967, which was known then as the "Pop Art Biennial" because of the participation of American artists from that movement. The other prize-winning artists were César from France, Fumiaki Fukita from Japan, David Lamelas from Argentina, Carlos Cruz Diez from Venezuela, Michaelangelo Pistoletto from Italy, Joshua Reichert from West Germany and Jan Schoonhoven from Holland. Other Polish artists exhibited at the Biennale included sculptors Jerzy Bereś and Jerzy Jarnuszkiewicz and graphic artist Lucjan Malinowski. Among the 14 photographers from Poland presented at the Biennial, particularly noteworthy was the work of the Polish visual artist Natalia Lach Lachowicz (Nona Bienal De São Paulo 1967).

2    The exhibition of Polish art was also announced in a note published in the magazine *Primera Plana* in August 1967: "Arte actual de Polonia: Una muestra seleccionada por el crítico Ryszard Stanislawski, que aporta una muestra de la pintura, la tapiceria y el arte grafico polacos (Instituto Di Tella, Florida 936)" (contemporary Polish Art: An exhibition selected by the critic Ryszard Stanisławski, which provides a sample of Polish painting, tapestry and graphic art (Instituto Di Tella, Florida 936)). (*Primera Plana* 1965, unnumbered page).

3    The exhibition was held at the Staatlichen Kunsthalle in Baden-Baden from 30 January to 9 May 1965.

4    El "Süddeutsche Zeitung" escribe al respecto [de la muestra Baden-Baden]: "Mientras que en algunos países del bloque del Este (Polonia y Checoslovaquia), el teatro ha recibido un fuerte impulso evocador y sugeridor por parte de pintores y decoradores cuyo ingenio corre parejo con su fantasía, en otros países no se ha producido tal evolución. Los teatros de la República Federal ocupan una posición intermedia entre el radicalismo de los jóvenes, como Tadeusz Kantor que en su 'Manifiesto pae el Teatro Cero', rechazan vehemente todas las direcciones estilísticas vigentes en el teatro actual, y el punto de vista conservador; la escenografía es la 'ancilla dramiatis', no cumpleindo más que con una misión de intermediaria".

5    bezinteresowne stany.

6    zanik woli apatia, znudzenie, monotonia.

7    For an English translation of Kantor's texts see: Kobialka (1993).

8    Kantor's plays were presented in cities such as Mexico City, where he presented *La clase muerta (Umarla klasa—The Dead Class)* from 2 to 10 March 1979. At the Teatro Galeón de Caracas, where he also presented *The Dead Class* in July 1978 as part of the IV World Session of the Teatro de las Naciones, and *Wielopole, Wielopole* from 28 July to 2 August 1981 at the Sala José Félix Ribas. In Guanajuato he presented *Wielopole, Wielopole* from 28 April to 1 May 1982 at the Teatro Principal, as part of the Festival Internacional Cervantino. After: Chrobak et al. (2000, pp. 119, 123, 124, 130, 144).

9    A translation of Tadeusz Kantor's *Teart Śmierci (The Theatre of Death)* was published in Buenos Aires in 1984. In 1986 the script of *Wielopole, Wielopole* was also published in Argentina and, in 1991, Kantor's self-analytical article on his position with regard to the theatrical avant-garde, entitled "Stanislavsky, Meyerhold y yo" (Stanislavski, Meyerhold and I). See: (Kantor 1984a), (Kantor 1975), (Kantor 1986, pp. 123–33) and (Kantor 1991, pp. 7–11). There is also a book by Marcos Rosenzvaig (Rosenzvaig 2008) *Tadeusz Kantor o los espejos de la muerte (Tadeusz Kantor. The Mirrors of Death)*, which is a synthesis of the biography and artistic and theatrical work of Tadeusz Kantor. Articles on Kantor were also published in Argentinean theatre magazines such as Schóó (1984, pp. 26–30), Cosentino (1987, pp. 64–70) and Kantor and Mantini (1997).

10   According to an interview with Carlos Fos conducted in Buenos Aires on 24 September 2014.

11   "revolución escénica". Ibid. On the subject of Kantor's theatre's reception in Latin America and his dialogue with Latiun American stage, see Parola Leconte (2006, pp. 287–94, 291–92).

12   La visita de Tadeusz Kantor y su Teatr Cricot 2 de Cracovia es, junto con la de Pina Bausch, la más trascendente de cuantas haya recibido el San Martín en toda su historia. Y el espectáculo que presenta, Wielopole Wielopole, una experiencia estética y humana verdaderamente singular, cuya influencia en la escena porteña dejaría una huella persistente.

13   La labor de Tadeusz Kantor, el director y artista plástico polaco, resulta, para muchos realizadores de esta tendencia [del teatro de imagen], el paradigma. Las obras de este realizador que se dieron en Buenos Aires, Wielopole Wielopole y ¡Que revienten los artistas!, así como sus escritos reunidos en el libro El teatro de la muerte impresionaron vivamente a algunos actores y directores que vieron allí un camino a seguir. Otros artistas siguieron una línea imprecisa entre teatro y performance.

14   Sagaseta refers to Argentinean theatre groups such as Alberto Félix Alberto and La Organizacion Negra, and directors such as Javier Margulis, who could have directlley entered into dialogue with Kantor's theatre.

15  Among the numerous articles that appeared in the Argentine press after Kantor's two visits to Buenos Aires, the most notable were: Mazas (1984). Berruti (1984, pp. 1–2). Ure (1984, p. 8). *Tiempo Argentino* (1984, pp. 1–5). Tirri (1984, p. 2). Cosentino (1987). Mazas (1987). Berruti (1987, p. 3). Lionetti (1987, pp. 34–36).

16  According to a telephone interview with Ana Alvarado in Buenos Aires on 27 September 2014.

17  All translations from the Polish are by the author.

18  Kantor's objects are different from the everyday objects used, for instance, by Kurt Schwitters, a representative of German Dadaism. In his collages and spatial compositions, called *Merz* and produced from 1919 onwards, Schwitters introduced extra-artistic components: pieces of wood, industrial elements, fragments of newspapers and also rubbish. He particularly exploited their *unfertig* aspect: never ready and never finished, rough, perishable and circumstantial. On this subject, see: Burns Gamard (2000) and Erdelfield (1992).

19  . . .najprostszy, ze śladami zużycia, wytarty przez długie używanie, u progu śmietnika. Już przez to samo: nieprzydatny życiowo, bez nadziei spełnienia swojej życiowej funkcji. Bez wartości praktycznej. Stary grat! Po prostu: biedny!

20  "[un] objeto real, físico, artificial, irracional, encontrado, construido, perturbado o interpretado [que] es sometido a una acción".

21  It can be seen that the use of the object in the theatre of the Periférico de Objetos also showed the influence of Ariel Bufano, an Argentinean theater director and from 1977 head of El Grupo de Titiriteros del Teatro San Martín (The Puppet Theatre of the San Martin Theatre) in Buenos Aires; he redefined puppet theatre in the country.

22  "A nosotros nos interesa incorporar en la escena el objeto tomado y no modificado, usado artísticamente. El objeto al natural aporta cierta genuinidad".

23  See Barthes (1968); also Barthes et al. (1982, pp. 81–90). Translations from the French are by the author.

24  détails concrets.

25  pris à vif.

26  Inutiles.

27  superflus par rapport à la structure [semiotique].

28  vraisemblable.

29  nous sommes le réel.

30  "de véritables organes de la vie psychologique secrète [. . .] [et qui] ont, comme nous, par nous, et pour nous, une intimité.

31  During his career, Kantor made numerous trips to Paris, the first of which was in 1947, through a grant from the Polish Ministry of Culture. The artist spoke French and had contacts in the theatre scene of the French capital.

32  "Non que je veuille faire de l'objet un acteur: je veux faire vivre l'objet, pour que l'objet devienne comme un organisme vivant. . ."

33  En el teatro de objetos actual, el objeto no remplaza al actor [. . .] El objeto debe estar en escena cuando su cosidad viviente es significante, cuando su presencia estructura no sólo la forma sino también el sentido de la obra.

34  On this subject, see: Halczak and Renczyński (2007).

35  Los personajes de Kantor no están ni vivos ni muertos. Fantasmas del pasado o habitantes de los sueños del proprio Kantor son convocados por éste en la escena. Con su cuasi vida repiten incesantemente algo o entran y salen de escena reiteradamente. A veces se confunden con maniquíes o con otros objetos de la escena, otras veces van adheridos a los objetos que definen: Mujer detras de la Ventana (Ella misma porta la ventana a través de la cual mira). Ella solo existe en forma concreta porque está unida a la ventana. El elenco de actores de Kantor está compuesto de "errantes eternos", los cuerpos y trajes, "embalajes" de varias capas, están soldados, pegados o cosidos a valijas, bicicletas, ventanas, guillotinas.

36  "el actor no manipulador".

37  "se vincula con el objeto sin convertirse en el".

38  Kantor used mannequins for the first time in 1967 in this play *Kurka wodna* (The Water Hen).

39  "objetos antropomórficos".

40  "sujetos escénicos".

41  "acciona sobre [el] objeto de tal modo que no es posible asegurar donde termina uno y comienza otro".

42  Materię spektaklu tworzyło „wewnętrzne życie" PRZEDMIOTU, jego właściwości, przeznaczenie, jego obszar imaginacyjny. Aktorzy stawali się jego żywymi częściami, organami. Byli jakby genetycznie z tym przedmiotem złączeni. Tworzyli jakiś BIO-OBIEKT działający, wydzielający tkankę dość szczególnej akcji.

43  "también ser 'el otro' que se encarna en su cuerpo humano".

44  "no manipulador".

45  His acceptance in the early 1980s, i.e., in the context of Polish Solidarity and Martial Law (from December 1981 until July 1983) of two offical prizes was very controversial. In 1981 Kantor was awarded the Prize of the Ministry of Culture and Art for his work as a stage designer, and in 1982 he received a Diploma of the Ministry of Foreign Affairs for the promotion of Polish culture abroad.

46  The play, written by Witkiewicz in 1921, was edited after World War II, in 1948 (Witkiewicz [1921] 1974).

47  About this subject, see: Schorlemmer (2007).

48    Sí, en dos ocasiones. La primera fue en 1980, cuando surgía el movi miento Solidaridad. En ese sentido, creo que mi obra predijo todo lo que pasó con la revolución de Lech Walesa [Wałęsa] y la Guerra que vino después. En esa época, hicimos representaciones en los astilleros navales de Gdansk [Gdańsk], allí donde empezó el sindicato Solidaridad. La segunda vez fue en diciembre de 1983, en pleno estado de sitio.

49    La reacción del público fue muy especial. La temperatura era mayor; pero no se si eso es bueno porque no me gusta demasiado cuando la obra de arte suscita una reacción en la cual es la política la que eleva la temperatura. Es la obra de arte la que debe crear una temperatura especial. [...] En realidad, yo estoy en contra del teatro político.

50    "en Polonia, en [ese] momento, [había] muchos directores que se [aprovechaban] de la política, pero sólo para tener éxito, para hacer carrera".

51    Kantor is probably referring here to Jerzy Grotowski, another world-renowned Polish theatre director of that time, who during the proclamation of the Martial law in December 1981, was in Italy and who remained abroad—he moved to the United States—and openly criticized the Polish govermnet.

52    "el teatro que [hacía] el Cricot 2 no [era] teatro político, no [tenía] ningún programa político que lo [sustentara]. Aunque, en sus capas más profundas, [tuviera] una significación política".

53    *Der Sandmann* (The Sandman), was part of E.T.A. Hoffmann's *Night Tales*, from 1817 (Hoffmann 1817).

54    Desde el punto de vista semántico, Veronese construye en Cámara Gesell [el cuarto espectáculo realizado por el Periférico de Objetos, en febrero de 1994] una imagen del hombre absolutamente violenta, pesimista y negativa, elabora una antropología nihilista que se hace cargo de todas las tragedias del pasado (y de las previsibles del futuro) sin concesiones al conformismo ni a la Esperanza.

55    This stage began in 1975 with the manifiesto of the *Theatre of Death* (Teatr Śmierci) and with the play *The Dead Class (Umarła Klasa)*. It continued until Kantor's death in 1991 with four more plays: *Wielopole, Wielopole* (1980), *¡Qué revienten los artistas! (Niech szczezną artysci!—Let the artists die!)* (1985), *Nunca más volveré aquí (Nigdy tu już nie powrócę—I Shall Never Return)* (1988) and *Hoy es mi cumpleaños (Dziś są moje urodziny—Today is my birthday)* (1991). About this subject see: Pleśniarowicz (1990). Kobialka (2009).

56    sprowadzona w okolice zera.

57    jest tak szczupła i mizerna, że aktorzy muszą walczyć, żeby się na niej utrzymać.

58    "no [dejaba] indiferente al espectador" sino que lo interpelaba con una "estética perturbadora".

59    "público ominosamente pegado a las escenas más crueles y obscenas".

60    Martin Rodríguez, in his article "El teatro de la desintegración" (The Theatre of desintegration) (Rodríguez 1999, p. 5) compares *La Cámara Gesell* with the threat theatre of Harold Pinter.

61    una estética de lo obsceno porque uno de los elementos periféricos es la obscenidad: mostrar lo que la gente no está esperando ver, hacer visible lo que debería haber quedado fuera de escena.

62    Freud's 1919 essay *Das Unheimliche* (Freud [1919] 1989), known in Spanish as *Lo siniestro (The Sinister)*, has been canonically translated into English as *The Uncanny*, and the term functions as such in literary and psychoanalytical studies. The adjective "unheimliche", however, can also be translated into English as "sinister", such as in the title used for the 1965 German film *Der unheimliche Mönch* in its U.S. release—*The Sinister Monk*.

63    "fuertemente basado en la noción freudiana de *lo siniestro*".

64    Los desaparecidos están muertos y lo siniestro es sacar el tema a la luz. Es tan siniestro lo que occurió que la Sociedad prefiere que eso quede tapado, enterrado. "El periférico de objetos", al sacar este tema a la luz se transforman en obscenos. Muestran lo que la gente no quiere ver, y eso sólo lo pueden hacer por medio de muñecos. Aquí llegamos a la esencia periférica: la elección de trabajar con muñecos—que ya tienen una carga siniestra—está relacionada con la posibilidad de contar historias horribles que no se podrían relatar con actores.

65    Si bien había en todos nosotros una necesidad política personal de exorcizar ciertos temas relacionados con la represión militar, ésta no estuvo presente a la hora de preparar el trabajo. Pero obviamente en nuestro país la obra fue leída como una obra sobre los desaparecidos por la dictadura. Nosotros, sin habernos propuesto hablar específicamente del tema, logramos una síntesis poética creo que imposible de lograr si la idea hubiera estado delante de la forma. Simplemente dejamos vagar nuestros fantasmas sobre lo siniestro. Y es imposible que en la Argentina determinados signos no se lean de esa manera; siendo lo siniestro un elemento con el cual convivimos durante años. El resultado no fue panfletario. Ese fue, creo, el mayor logro del trabajo.

66    Según palabras del público el resultado era aterrador. La gente no podía soportar verlo, pero al mismo tiempo no dejaba de verlo. Como si, a pesar de la necesidad de salir corriendo de la sala, una fuerza los mantenía sujetos a la platea. Poder animar algo que no tiene vida produce esa fascinación, creo, tan difícil de igualar. Es conocer algo sobre la muerte. El público necesita ver algo sobre la muerte aunque lo rechace.

67    "mroczne pochody makabrycznych lalek".

68    On this subject, see Kristeva (1980). An interpretation of Tadeusz Kantor's theatre work in relation to the concept of the abject can be found in an article by Watt (2014, pp. 52–62). Óscar Cornago proposes an analysis of the work of Tadeusz Kantor and other theatre makers of the latter half of the twentieth century, based on the theory of Jean-François Lyotard, in his *Dispositivos libidinales* from 1975, or *Economía libidinal* from 1980 (Cornago 2005, pp. 63–74).

[69]  "en el muerto nos podemos reconocer, en el sentido de que el muerto es de nuestra misma especie, es un ser humano como nosotros. Pero al mismo tiempo es un extraño, es alguien que esta fuera de vida".

[70]  Also cited by De la Torre (2005, p. 342).

[71]  „MANEKIN w moim teatrze ma stać się MODELEM, przez który przechodzi silne odczucie ŚMIERCI i kondycji UMARŁYCH. Modelem dla ŻYWEGO AKTORA".

[72]  "remiten de manera particular al tiempo y a la muerte".

[73]  In the words of Ana Alvarado in an interview María Castillo conducted with her and Daniel Veronese.

[74]  Jego pojawienie się [manekina w moim teatrze] zgadza się z moim przekonaniem coraz mocniejszym, że życie można wyrazić w sztuce jedynie przez brak życia, przez odwołanie się do ŚMIERCI, przez POZORY, przez PUSTKĘ i brak PRZEKAZU.

[75]  En sus obras, [las troupes de teatro del periodo de la post-dictadura en Argentina] se refieren no sólo a esta etapa de la historia Argentina: Resistencia, desaparición, violencia, tortura, represión, lo siniestro, etc., sino también a la crisis social y económica del periodo posterior, encarada en la escena en varios sentidos: la parodia, el absurdo, el sinsentido y lo patetico. Esos aspectos no son "temas" ni "influencias" del context social en la obra sino que son las piezas que construyen la propia obra y los procedimientos de los artistas.

[76]  On the subject of the theatre in Argentina in the context of the post-dictatorship era, see also Persino (2003, pp. 236–42).

[77]  On this issue, see Longoni and López (2012, pp. 128–30).

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
