# Peer review of "Dialogue between the Concept of the Object in the Theater of Tadeusz Kantor and the Theatrical Praxis of the Periférico de Objetos"

_arts, 2023_

Round 1

Reviewer 1 Report

Comments and Suggestions for Authors

The article addresses the Polish artist T. Kantor and his influences on Argentinean art world, primarily theater. The research issue, argumentation and conclusion are all very interesting and important. I do have some comments:

- Related to unclear theoretical background. The author discusses Kantor's understanding of the 'poor' objects and their agency in theater and at art exhibitions. Although very interesting, I miss the use of literature on affect and atmosphere, which (a) gives a broader insight into the agency of objects (e. g. Gell and his sucessors, maybe even Harman's discussion on object-oriented ontology); and (b) provides understanding of the affect and capacity of (artists and audience) of being affected. That might provide better understanding of how Kantor's work was in Argentina interpreted through the experiences of violence of the dictatorship, even though violence was not of interest to Kantor (p.10).

- I suggest clarification of methods in the introduction as methodology of the research is not clear.

In general, I find the article very interesting.

Comments on the Quality of English Language

In general, the language is clear, but could use a proofreading

- Quotation in non-English language is somehow obtrusive to reading Sometimes it comes second, sometimes first.... I don't think it is necessary to keep both versions at all, but if quotes are kept in two languages, I suggest always adding the original non-English quote in brackets or footnotes.

- Exorcise: on several occasions the author uses the word exorcise. Does (s)he mean exercise (as e.g. agency, action...)?

Author Response

Response to Reviewer 2 Comments

1. Summary

Thank you very much for taking the time to review this manuscript. I am truly honored and happy that you find the article valuable for the publication with some improvements. Thank you for your time and attention. I would like to thank you especially for your comment about the theoretical background.

I am not a specialist of the theory of affect and atmosphere, but I will try to follow your suggestions and to develop this part of my text.    

I will as well try to improve the quality of English, as suggested.

 As suggested, I will put non- English quotations in the footnotes.

I was using the word exorcise in order to describe a king of ritual of exorcism, very present in Kantar’s oeuvre and that of the Periferico de Objetos.

Thank you for your work and help. I truly appreciate all comments.  

With my best regards.

Reviewer 2 Report

Comments and Suggestions for Authors

For a more readable reception, it is suggested to specify subheadings typical of a scientific text, e.g. research, results, conclusions, etc.

Comments on the Quality of English Language

Article has a lot of paragraphs in original spanish language, but is difficult to understand all. It is suggested to change the language to make the entire text legible.

Author Response

Response to Reviewer 2 Comments

1. Summary

Thank you very much for taking the time to review this manuscript. I am truly honored and happy that you find the article ready for the publication. Thank you for your time and attention. I would like to thank you especially for your comment about the Spanish language – I will put it in the footnotes. Thank you.

With my best regards.

Reviewer 3 Report

Comments and Suggestions for Authors

I consider the peer-reviewed article to be valuable and worthy of publication. The reception and perception of Kantor's theatre in Latin America, particularly in Argentina, has not been more widely discussed. Nor has there been any comparison of Kantor's theatre to theatres operating in Argentina that use puppets as a means of artistic expression. Therefore, the author of this article fills the gap in our knowledge of Kantor's oeuvre and activities and reconstructs the lesser-known context in which his art functioned.
Undoubtedly, the author is in control of the archival material, working it out well, and creating a clear and concrete narrative. He synthetically describes both Kantor's ideas and the artistic ideas of Argentinian artists. Importantly, he makes free use of both Polish and Argentine sources. The result is wholly satisfactory.
The only issue that caught my attention is the one related to Kantor's designation as an Eastern European artist. Undoubtedly, Kantor was an artist in one of the Eastern Bloc countries. Culturally, on the other hand, he was an artist derived from and conditioned by the history of Central Europe. We may say that he was a Central European artist.

Author Response

Response to Reviewer 3 Comments

1. Summary

Thank you very much for taking the time to review this manuscript. I am truly honored and happy that you find the article ready for the publication. Thank you for your time and attention. I would like to thank you especially for your comment about Kantor’s origin. I am myself from Poland and I am using my Eastern European identity, but maybe for the non-Polish audience it will be clearer to define Poland as Central Europe. Thank you for bringing this discussion here. I will reconsider this part of my article.

With my best regards.

Reviewer 4 Report

Comments and Suggestions for Authors

 This is a fascinating and interesting article, which shows how an apolitical theatrical style from communist Eastern Europe is transformed into a political theatrical style in Argentina that creates new meanings for Kantor's original concepts. 

Author Response

Response to Reviewer 4 Comments

1. Summary

Thank you very much for taking the time to review this manuscript. I am truly honored and happy that you find the article ready for the publication. Thank you for your time and attention. With my best regards.